# Poly(vinylidene fluoride) Composite Nanofibers Containing Polyhedral Oligomeric Silsesquioxane–Epigallocatechin Gallate Conjugate for Bone Tissue Regeneration

**DOI:** 10.3390/nano9020184

**Published:** 2019-02-01

**Authors:** Hyo-Geun Jeong, Yoon-Soo Han, Kyung-Hye Jung, Young-Jin Kim

**Affiliations:** 1Department of Biomedical Engineering, Daegu Catholic University, Gyeongsan 38430, Korea; jhg2833@empal.com; 2Department of Advanced Materials and Chemical Engineering, Daegu Catholic University, Gyeongsan 38430, Korea; yshancu@cu.ac.kr (Y.-S.H.); khjung@cu.ac.kr (K.-H.J.)

**Keywords:** bone regeneration, poly(vinylidene fluoride), composite nanofiber, piezoelectricity, antioxidant activity

## Abstract

To provide adequate conditions for the regeneration of damaged bone, it is necessary to develop piezoelectric porous membranes with antioxidant and anti-inflammatory activities. In this study, composite nanofibers comprising poly(vinylidene fluoride) (PVDF) and a polyhedral oligomeric silsesquioxane–epigallocatechin gallate (POSS–EGCG) conjugate were fabricated by electrospinning methods. The resulting composite nanofibers showed three-dimensionally interconnected porous structures. Their average diameters, ranging from 936 ± 223 nm to 1094 ± 394 nm, were hardly affected by the addition of the POSS–EGCG conjugate. On the other hand, the piezoelectric β-phase increased significantly from 77.4% to 88.1% after adding the POSS–EGCG conjugate. The mechanical strength of the composite nanofibers was ameliorated by the addition of the POSS–EGCG conjugate. The results of in vitro bioactivity tests exhibited that the proliferation and differentiation of osteoblasts (MC3T3-E1) on the nanofibers increased with the content of POSS–EGCG conjugate because of the improved piezoelectricity and antioxidant and anti-inflammatory properties of the nanofibers. All results could suggest that the PVDF composite nanofibers were effective for guided bone regeneration.

## 1. Introduction

A variety of techniques have been tried for the regeneration of bone defects, including bone grafting, distraction osteogenesis, and guided bone regeneration (GBR) [1,2,3]. Among them, GBR has been used as an efficient method for both the reconstruction of the structure of, and reestablishment of the function in, bone defects and damaged tissues. The GBR approach principally uses a membrane acting as a barrier membrane in between soft tissue and bone tissue defect. GBR membranes prevent the migration of faster-growing fibrous tissue into the defect and secure osteoconduction (bone growth) to promote bone regeneration [3]. Thus, GBR membranes need to have biocompatibility to support bone tissue reconstruction and mechanical stability to maintain a protective space during bone restoration process.

Nanofibrous membranes have structural similarity to the extracellular matrix (ECM) and high specific surface area [4]. GBR membranes composed of ultrafine nanofibers are more suitable for cell attachment and proliferation than conventional membranes with large scale structures. Various strategies—such as self-assembly, wet-spinning, and electrospinning—have been used to fabricate GBR membranes [4,5,6,7,8,9,10]. Of these strategies, electrospinning has attracted a great deal of attention because this method is very simple and versatile, and can produce various polymer nanofibers. Electrospun polymer nanofibers have unique advantages in the biomedical fields such as drug delivery and tissue engineering, as they can form porous structures that simulate the features of the ECM [2,4,11,12].

Poly(vinylidene fluoride) (PVDF) is an attractive semicrystalline polymers for use in biomedical applications because of its efficient piezoelectricity and biocompatibility [13,14]. The strong piezoelectric effect observed in PVDF is directly linked to the crystalline phase arrangement. PVDF exits in one of five polymorphic crystalline phases (α, β, γ, δ, and ε) dependent on its distinct chain conformation [15]. The nonpolar α-phase does not exhibit an efficient conformation for piezoelectric application, but it does present good mechanical properties and thermal stability. On the other hand, β- and γ-phases exhibit electroactive and polar properties. Among them, the β-phase has a higher density net dipole moment and exhibits more promising piezoelectric properties. Therefore, increasing the piezoelectric β-phase content in PVDF is very important for its biomedical application because piezoelectric materials can provide electrical stimulation to cells to promote tissue formation and can use as a scaffold for tissue engineering [14,16]. This phenomenon is very similar that electrically charged ECM stimulates the cell growth during tissue regeneration [14]. In particular, bone tissue and its constituent collagen fibers possess intrinsic piezoelectricity to play a significant role in regeneration of the damaged bone, and thus a variety of piezoelectric materials have been applied to promote bone regeneration both in vitro and in vivo [16,17].

Polyhedral oligomeric silsesquioxane (POSS) molecules have a unique nanostructure comprised of an inorganic silsesquioxane cage core and an organic functional group shell [18]. A variety of POSS molecules contain reactive functionalities, making them suitable precursors for grafting. Therefore, they have been used in various biomedical applications including tissue engineering due to their distinctive structure and excellent biocompatibility [19,20]. Epigallocatechin gallate (EGCG) is a polyphenolic flavonoid derived from a variety of plants and, in particular, is a major bioactive component in green tea, which has been shown to possess a variety of pharmacological functions; for instance, it exhibits antioxidant, anticancer, and anti-inflammatory properties [21,22,23]. EGCG inhibits lipopolysaccharide (LPS)-induced osteoclastic bone resorption and attenuates inflammatory bone loss in bone metabolism [24,25]. Moreover, EGCG increases osteoblast proliferation and alkaline phosphatase (ALP) activity, which leads to increased osteoblastic bone formation [26]. 

Piezoelectric materials can change their surface charge without external energy power source [14]. Several research papers have reported an observed enhancement of the piezoelectric β-phase content in PVDF fabricated by electrospinning with the incorporation of fillers, such as metal nanoparticles, ceramics, and inorganic salts [27,28,29,30]. In addition, EGCG-incorporated membranes have been fabricated and have been observed to promote osteoblastic proliferation and prevent inflammatory responses [26]. Therefore, in this study, we fabricated PVDF/POSS–EGCG conjugate composite nanofibers, which showed improved piezoelectric, antioxidant, and anti-inflammatory properties, to support bone tissue regeneration using an electrospinning method. We systematically examined the effect of the POSS–EGCG conjugate content on the structure and physicochemical properties of the composite nanofibers obtained. Furthermore, the in vitro bioactivity of the composite nanofibers was investigated through the cell proliferation, ALP activity, and bone mineralization assays.

## 2. Materials and Methods

### 2.1. Materials

PVDF (*M_w_* = 275000 g/mol), EGCG, lipopolysaccharide (LPS), horseradish peroxidase (HRP), hydrogen peroxide (H_2_O_2_, 30 wt% in H_2_O), dimethyl sulfoxide (DMSO), *N,N*-dimethylformamide (DMF), tetrahydrofuran (THF), and isopropyl alcohol (IPA) were obtained from Sigma-Aldrich Co. (St. Louis, MO, USA). Xanthine oxidase (XO from buttermilk, EC 1.1.3.22), xanthine, and nitro blue tetrazolium (NBT) were purchased from Wako Pure Chemical Industries (Osaka, Japan). Aminopropylisobutyl polyhedral oligomeric silsesquioxane (API-POSS) was purchased from Hybrid Plastics (Hattiesburg, MS, USA). Osteoblast-like cell line MC3T3-E1 derived from mouse calvaria was acquired from the American Type Culture Collection (Manassas, VA, UAS) and a murine macrophage cell line RAW 264.7 was purchased from the Korean Cell Line Bank (Seoul, South Korea). All materials were used as received without further purification.

### 2.2. Synthesis of the POSS–EGCG Conjugate

The POSS–EGCG conjugate was synthesized as follows. API-POSS (174.9 mg, 0.2 mmol) and EGCG (137.5 mg, 0.3 mmol) were first dissolved in 30 mL of a mixed solvent of THF/IPA/deionized water (DW) at a ratio of 2:1:1 before adding a solution of HRP (5 mg, 750 units) dissolved in 3 mL of Dulbecco’s phosphate-buffered saline (DPBS, pH 7.4). To this resultant solution, 5 mL of 5 wt% H_2_O_2_ was added five times every 10 min. After 4 h, the resultant POSS–EGCG conjugate was dialyzed in DW for 48 h. Then, the precipitated conjugate was isolated by centrifugation and washing with DW in four replicates, followed by drying in vacuo. The amount of EGCG conjugated to API-POSS was determined by elemental analysis using a Flash 2000 elemental analyzer (Thermo Fisher Scientific, USA). The structures of EGCG, API-POSS, and the POSS–EGCG conjugate were determined by ^1^H NMR spectroscopy (AVANCE III 400, Bruker BioSpin, USA).

^1^H NMR of EGCG (400 MHz, D_2_O), δ (ppm) = 6.91 (CH of D-ring), 6.52 (CH of B-ring), 6.10, 6.07 (CH of A-ring), 5.48, 4.97 (CH of C-ring), 2.88 (CH_2_ of C-ring).

^1^H NMR of API-POSS (400 MHz, CDCl_3_), δ (ppm) = 2.83 (CH_2_NH), 1.84 (CHCH_3_), 1.58 (CH_2_CH_2_CH_2_), 0.95 (CHCH_3_), 0.56 (SiCH_2_).

^1^H NMR of POSS–EGCG (400 MHz, DMSO-d_6_), δ (ppm) = 6.81 (CH of D-ring), 6.41 (CH of B-ring), 5.95, 5.84 (CH of A-ring), 5.36, 4.98 (CH of C-ring), 2.77 (CH_2_ of C-ring) for EGCG, 2.69 (CH_2_NH), 1.82 (CHCH_3_), 1.58 (CH_2_CH_2_CH_2_), 0.95 (CHCH_3_), 0.54 (SiCH_2_) for API-POSS.

### 2.3. Fabrication of PVDF Composite Nanofibers 

Before electrospinning the PVDF composite nanofibers, the PVDF (1.8 g) was dissolved in 10 mL of a mixed solvent of DMF/THF in a ratio of 1:1 to obtain an 18 wt% solution. After stirring 12 h, the POSS–EGCG conjugate was added to the PVDF solution with a different weight ratio at 60 °C for 8 h. The contents of POSS–EGCG conjugate in solutions were 0, 2, 4, and 6 wt% with respect to the weight of the PVDF. The mixed solution of PVDF and the POSS–EGCG conjugate was filled into a 20 mL standard syringe with a 22 G stainless steel needle (internal diameter = 0.41 mm). Then, this solution was electrospun into nanofibers at room temperature with a feeding rate of 2 mL/h feed rate, a voltage of 18 kV, and a working distance of 15 cm. After electrospinning, the PVDF composite nanofibers were peeled off from the stainless steel plate and vacuum dried for 12 h.

### 2.4. Characterization of the PVDF Composite Nanofibers

The morphology of the PVDF composite nanofibers was visualized by scanning electron microscopy (SEM, S-4300, Hitachi, Japan) at an acceleration voltage of 5 kV after sputter coating of samples with gold. Image-Pro Plus software (Media Cybernetics Inc., Rockville, MD, USA) was used to determine the average fiber diameters for the composite nanofibers. Energy dispersive spectroscopy (EDS) spectra were obtained for the determination of the distribution profile of the POSS–EGCG conjugate on the nanofiber surface.

Fourier transform infrared spectroscopy (FTIR) measurement was performed using an ALPHA spectrometer (Bruker OpticsBillerica, MA, USA) with a resolution of 4 cm^−1^. Attenuated total reflectance (ATR) mode was used to analysis the characteristic bands over a range from 400 to 1600 cm^−1^. The relative β-phase content (F(β)) in the samples was determined from the following equation [29].
$$F\left(\beta \right)=\frac{A_{\beta }}{\left(\frac{K_{\beta }}{K_{\alpha }}\right)A_{\alpha }+A_{\beta }}$$
where A_α_ and A_β_ represent the absorbance of peaks related to the α- and β-phases at 760 and 840 cm^−1^, respectively, and K_α_ and K_β_ are the absorption coefficients at the respective wavenumbers, which are 7.7 × 10^4^ cm^2^/mol and 6.1 × 10^4^ cm^2^/mol.

X-ray diffraction (XRD) patterns were also used to evaluate the crystalline phases of the PVDF composite nanofibers and thus XRD measurements were performed using a Rigaku D/MAX-2500V/PC X-ray diffractometer (Japan) with high intensity Cu Kα radiation at 40 kV/100 mA. The diffractograms were scanned in a 2θ range from 10° to 50°. The composite nanofiber samples were analyzed using X-ray photoelectron spectroscopy (XPS) to obtain the surface elemental composition. XPS measurements were performed using a Quantera SXM (ULVAC-PHI Inc., Japan) equipped with a monochromatic Al Kα X-ray source (1486 eV). The photoelectron take-off angle was fixed at 45° relative to the sample surface. Besides XPS measurements, static contact angles were measured with a DSA 100 contact angle meter (KRÜSS, Germany) to examine the hydrophobicity of the composite nanofiber. Each contact angle of the sample is an average of five measurements. The mechanical properties of the composite nanofibers were investigated with a TO-101 universal testing machine (Testone Co., Siheung, South Korea) with 2 kN load capacity at a rate of 10 mm/min. All samples were cut into rectangular specimens with a size of 3 cm × 1 cm and tested five parallel measurements.

### 2.5. Cell Proliferation

MC3T3-E1 cells were used to evaluate the in vitro bioactivity of the PVDF composite nanofibers by a cell proliferation assay in alpha minimum essential medium (α-MEM) containing 10% of fetal bovine serum (FBS) and 1% of penicillin–streptomycin at 37 °C with 5% CO_2_. Before assaying, the composite nanofibers were cut into circular shape with a diameter of 15 mm, followed by putting into 24-well tissue culture plate and fixing with glass ring (inner diameter = 11 mm). Then, they were sterilized with 70% ethanol and rinsed with DPBS and α-MEM. After drying, the composite nanofibers were once again sterilized under UV radiation for 3 h.

The proliferation of viable cells on the composite nanofibers was determined using the 3-(4,5-dimethylthiazol-2-yl)-2,5-diphenyltetrazoliumbromide (MTT) assay. One milliliter of cell suspension (2 × 10^4^ cells/well) was placed onto the sterilized samples and cells were cultured for different periods of time. After culturing for 1, 3, 5, 7, and 14 days, culture media were replaced with 0.2 mL of the MTT solution and further incubation of the cells was maintained for another 4 h. Next, we removed the remaining media and added 1 mL of DMSO to solubilize the precipitated formazan crystals. Finally, the resulting supernatant was transferred to 96-well plate with 0.2 mL per well. The absorbance was determined at 570 nm using a spectrophotometric plate reader (OPSYS-MR, Dynex Technology Inc., USA). Furthermore, after days 3 and 7 of cell culture, the cell-nanofibers were fixed in 4% glutaraldehyde for 1 h, and then dehydrated using different concentrations of ethanol (25, 50, 70, and 100%), followed by vacuum-drying. The morphologies of dried samples were observed using SEM after sputter coating with gold.

### 2.6. Superoxide Anion Radical Scavenging Capacity Assay

Superoxide anion radicals were generated with xanthine/XO and determined as described in a previous paper of ours [31]. EGCG and the POSS–EGCG conjugate were first dissolved in the mixed solvent of DMSO and DPBS (pH 7.4) and diluted with DPBS. To measure the superoxide anion radical scavenging capacity, xanthine (30 μg/mL) was added to DPBS containing ethylenediaminetetraacetic acid (EDTA, 15 μg/mL), XO (40 milliunits/mL), NBT (130 μg/mL), and various concentrations of test sample. Changes in the absorbance at 560 nm over 10 min were recorded as a measure of the changing number of superoxide anion radicals.

The superoxide anion radical scavenging capacity of samples was determined according to the following formula.
$$\mathrm{Superoxide}\;\mathrm{anion}\;\mathrm{radical}\;\mathrm{scavenging}\;\mathrm{capacity}\;\left(\%\right)=\frac{{\mathrm{Abs}}_{\mathrm{control}}-{\mathrm{Abs}}_{\mathrm{sample}}}{{\mathrm{Abs}}_{\mathrm{control}}}\times 100$$
where Abs_control_ is the absorbance of control in the absence of sample and Abs_sample_ is the absorbance in the presence of samples.

### 2.7. ALP Activity and Bone Mineralization Assay

ALP activity can provide a useful index of the osteoblastic phenotype. Thus, the osteoblastic differentiation of MC3T3-E1 cells was assessed by measuring the ALP activity with an ALP assay kit (DALP-250, BioAssay Systems, USA) at periods of 3, 5, 7, and 14 days after cell seeding. The composite nanofibers seeded with cells washed with DPBS and incubated in 0.5 mL of DW containing 0.2% Triton X-100. The cell lysates were mixed with *p*-nitrophenyl phosphate solution. In the presence of ALP, *p*-nitrophenyl phosphate can be hydrolyzed to *p*-nitrophenol and the rate of *p*-nitrophenol production is proportional to the ALP activity. Therefore, the level of *p*-nitrophenol production was determined by measuring the absorbance at 405 nm and normalized to 1 × 10^4^ cells.

Bone mineralization capability of the composite nanofibers was assayed by the alizarin red S (ARS) staining method. ARS can selectively bind to calcium ions and thus calcium deposition is easily measured by the use of ARS. After 3, 5, 7, and 14 days of cell culture on the different nanofibers, the cells were fixed with 4% glutaraldehyde for 30 min and then stained with 1 mL of 40 mM ARS (pH 4.1). After incubation for 20 min, the specimens were washed thrice with DW to remove unreacted ARS and dissolved in 1 mL of 10% cetylpyridinium chloride. The absorbance of the supernatant at 540 nm was obtained using microplate reader and was normalized to 1 × 10^4^ cells.

### 2.8. Quantification of Inflammatory Cytokine

The effect of the PVDF composite nanofibers on the expression of the inflammatory cytokine interleukin-6 (IL-6) was measured using the RAW 264.7 cells. For the quantification of IL-6 production, the cells (2 × 10^4^ cells/well) were precultured in Dulbecco’s modified Eagle’s medium (DMEM) containing 10% heat-inactivated FBS, 100 U/mL of penicillin, and 100 μg/mL of streptomycin for 24 h at 37 °C before adding 1 μg/mL of LPS into the cell culture plate. After activation of the cells for 24 h, the amount of IL-6 released into the media was measured by mouse IL-6 enzyme-linked immunosorbent assay kit (R&D Systems, USA) according to the manufacturer’s protocol.

### 2.9. Statistical Analysis

All data were represented as mean value ± standard deviation. Differences between two groups were analyzed with a one-way ANOVA followed by a Turkey test using SigmaPlot 13.0 (Systat Software, CA). A significant difference was defined for values of *p** ˂ 0.05.

## 3. Results and Discussion

### 3.1. Synthesis of the POSS–EGCG Conjugate

To prepare the POSS–EGCG conjugate, EGCG was conjugated to API-POSS via a one-step reaction using HRP as a catalyst (Figure 1). The structure of the resultant POSS–EGCG conjugate was characterized by ^1^H NMR spectroscopy (Appendix A). The characteristic signals belonging to EGCG units were observed at 6.81, 6.41, 5.95, 5.84, 5.36, 4.98, and 2.77 ppm. Moreover, the peaks due to API-POSS units in the POSS–EGCG conjugate were shown at 2.69, 1.82, 1.58, 0.95, and 0.54 ppm [20,32]. The semiquinone radicals or active oxygen species can easily oxidize either the gallyl moiety (B-ring) or the gallate moiety (D-ring) of EGCG, resulting to transform EGCG into reactive species with a quinone structure [33]. However, gallyl structures are more susceptible to oxidation than gallate structures and they only form catechol quinones through intermediate semiquinones [34]. In addition, the presence of a reactive amino group on API-POSS can provide a site for enzymatic conjugation of catechol through electrophilic addition [35]. The conjugation ratio of EGCG that was introduced to API-POSS was calculated from the result of elemental analysis to be 0.96.

### 3.2. Fabrication of the PVDF Composite Nanofibers

Composite nanofibers have been proven to act as effective mechanical supports and to promote osteoconduction in bone tissue regeneration [2,4]. In the present study, composite nanofibers were fabricated by electrospinning PVDF solutions containing different amounts of POSS–EGCG conjugate. The contents of POSS–EGCG conjugate in the mixed solutions were 0 (PVDF), 2 (PE02), 4 (PE04), and 6 wt% (PE06) with respect to the weight of the PVDF. In addition, the PVDF nanofiber containing 6 wt% of pure API-POSS (PO06) was also prepared as a control. The SEM observations revealed that thoroughly interconnected porous structures formed between the composite nanofibers (Figure 2). The average fiber diameter of the composite nanofibers, which was 1033 ± 270 nm for PVDF, 971 ± 262 nm for PE02, 936 ± 223 nm for PE04, 1094 ± 394 nm for PE06, and 1131 ± 281 nm for PO06, was hardly affected by adding the POSS–EGCG conjugate. In addition, the distribution of the POSS–EGCG conjugate in the PVDF composite nanofibers was observed using EDS. The EDS Si-mapping analyses of the composite nanofibers represented that the POSS–EGCG conjugate was homogenously distributed over the nanofibers and that a greater density of Si was detected on the PE06 nanofiber (Appendix A).

### 3.3. Physicochemical Properties of the PVDF Composite Nanofibers

The FTIR was used to analyze the chemical structure and phase composition of the PVDF composite nanofibers. Characteristic bands appeared at 1402 and 840 cm^−1^ in the spectrum of the pure PVDF nanofiber corresponded to the stretching and rocking vibration of the CH_2_ groups, respectively (Figure 3a) [28,29,36]. The four bands observed at 1275, 1178, 760, and 482 cm^−1^ corresponded to the deformation, stretching, bending, and wagging modes of the CF_2_ groups in PVDF. An absorption peak due to the skeletal vibration of C–C bonds in PVDF was also observed at 880 cm^−1^. After adding the POSS–EGCG conjugate or pure API-POSS, the PVDF composite nanofibers exhibited a new absorption peak at 1110 cm^−1^ associated with the stretching mode of Si–O–Si bonds, which is the typical absorption peak of the POSS inorganic framework [37]. In particular, the absorption peaks at 1402, 1275, 880, 840, and 482 cm^−1^ represent characteristic vibration modes of the piezoelectric β-phase of PVDF [28,29]. 

The FTIR data was used to calculate the relative β-phase content (F(β)) in the PVDF composite nanofibers. As presented in Figure 3b, the F(β) values of all samples were higher than 75% because the crystalline phase change of the PVDF from α-phase to β-phase was occurred by the electrostatic forces applied to the polymer droplet during the electrospinning [30]. In addition, the piezoelectric β-phase increased from 77.4 ± 2.5% to 88.1 ± 3.2% after adding the POSS–EGCG conjugate. According to previous reports, converting α-phase to piezoelectric β-phase in PVDF/ceramic composites was ascribed to the interaction between the negatively charged ceramic nanoparticles and the positively charged CH_2_ groups of PVDF [27,28]. Moreover, hydroxylated ceramic nanoparticles interacted more strongly with PVDF through the hydrogen bonds formed by the fluorine atoms on the PVDF with the hydroxyl groups on the surface of the ceramic nanoparticles. Therefore, we may deduce that, in our experiments, PE06 exhibited a higher F(β) value than those of the other composite nanofibers because of strong dipole and hydrogen bonding interactions between the PVDF and POSS–EGCG conjugates.

The surface elemental composition of the PVDF composite nanofibers was assessed using XPS. The XPS survey scan spectrum of the pure PVDF nanofiber exhibited two separate peaks which corresponded to the F 1s orbital (688 eV) and the C 1s orbital (287 eV) (Figure 4a). After adding the POSS–EGCG conjugate or pure API-POSS, the XPS spectra exhibited six separate peaks assigned to PVDF, the POSS–EGCG conjugate, and API-POSS: F 1s (688 eV), O 1s (535 eV), N 1s (400 eV), C 1s (287 eV), Si 2s (155 eV), and Si 2p (105 eV). The intensities of the peaks assigned to the POSS–EGCG conjugate increased with increasing contents of the POSS–EGCG conjugate. In addition, the crystalline phases of the PVDF composite nanofibers were examined by means of XRD as shown in Figure 5. All samples exhibited two diffraction peaks mainly attributed to the piezoelectric β-phase of PVDF at 20.4° and 35.9°, which were indexed to the β(110)/(200) and β(020)/(101) planes, respectively [27,28].

The investigation on the surface hydrophilicity of biomedical membranes is very important to confirm the affinity between and the membranes and the cell. Suitable hydrophilicity can influence the attachment and proliferation of the cells on the surface of the membranes. Therefore, the performance of membranes in biomedical applications may depend on the hydrophilicity [38]. Here we measured the water contact angle to evaluate the hydrophilicity of the composite nanofibers. Figure 6 exhibited the water contact angle data measured for the composite nanofibers; the PVDF nanofiber revealed a high water contact angle (127.7 ± 9.5°) due to its hydrophobic property. In addition, the water contact angles were hardly changed by the addition of the POSS–EGCG conjugate, i.e., 129.0 ± 8.7° for PE02, 128.9 ± 6.8° for PE04, and 130.4 ± 8.6° for PE06.

GBR membrane should be strong enough to withstand the force during new bone tissue regeneration. Thus, the effect of the addition of the POSS–EGCG conjugate on the mechanical properties of the composite nanofibers was assayed. As shown in Table 1, the resulting data obtained on the basis of stress–strain measurements exhibited that the tensile strength and Young’s modulus increased with increasing contents of the POSS–EGCG conjugate. This means that the POSS–EGCG conjugate can cause temporary intermolecular cross-linking of polymer chains and thus can provide enhanced mechanical strength [39]. However, the addition of POSS–EGCG conjugate induced a slight decrease in elongation at break of the composite nanofibers.

### 3.4. Cell Proliferation

The in vitro bioactivity of the PVDF composite nanofibers was evaluated to assess the potential of the nanofibers as GBR membranes. The cell proliferation on the PVDF composite nanofibers was studied by the MTT assay, resulting that the relative cell viability of the MC3T3-E1 cells cultured on the composite nanofibers indicated good cytocompatibility of the nanofibers (Figure 7). For all the tested nanofibers, time-dependent actions were observed in cell proliferation. Changes in the content of the POSS–EGCG conjugate in the composite nanofibers affected the cellular activities. All the composite nanofibers accelerated more rapid cell proliferation as compared with the pure PVDF nanofiber. Among the composite nanofibers, cell proliferation was significantly enhanced on PE06 in comparison with on the other nanofibers. These results were associated with the contents of piezoelectric β-phase and POSS–EGCG conjugate in the PVDF composite nanofibers.

During tissue regeneration, piezoelectricity can induce an increase in the surface charge density of ECM materials for the delivery of an electrical stimulus without external energy sources and consequently causes higher levels of cell proliferation and differentiation [14,16]. In addition, EGCG causes a significant elevation of osteoblastic survival as well as a decrease of osteoblastic apoptosis caused by reactive oxygen species, resulting that the proliferation and differentiation of osteoblasts can be stimulated because of antioxidant and free-radical scavenging activities of EGCG [26,40]. As shown in Figure 8, the POSS–EGCG conjugate exhibited excellent superoxide anion radical scavenging activity similar to that of pure EGCG, which could have a positive effect on cell viability. Therefore, the piezoelectric property and antioxidant activity of PVDF composite nanofibers may play an important role in osteoblast cell proliferation.

To confirm cell growth on the PVDF composite nanofibers, the interactions between the cells and nanofibers was observed using SEM on periods of 3 and 7 days after cell seeding. SEM images exhibited that the cells adhered and spread on the surface of the composite nanofibers, and cell proliferation increased time-dependently in all the tested nanofibers (Figure 9). This indicates excellent biocompatibility of the composite nanofibers. Moreover, almost the whole surface of the PE06 was covered by the cells at day 7 of cell culture.

### 3.5. ALP Activity and Bone Mineralization

ALP catalyzes the cleavage of organic phosphate esters and plays an important role during the formation process of bone nodule [1,2]. ALP activity is considered as an early marker of osteoblastic differentiation because it significantly increases during the differentiation step. Therefore, we examined ALP activity to evaluate the capability of the PVDF composite nanofibers for promoting osteoblastic differentiation. As a result, the cells cultured on PE06 exhibited higher ALP activity than those cultured on the other composite nanofibers at all time points (Figure 10a). Similar results have been previously reported, in that various flavonoids—such as green tea catechins, quercetin, and kaempferol—significantly increased ALP activity by the activation of the extracellular signal-regulated kinase pathway and endoplasmic reticulum binding [40,41,42].

The mineralized bone nodules are usually formed via three main stages such as cell proliferation, ECM development, and mineralization. Therefore, the osteogenesis capability of the cells as a late-stage marker for osteogenic differentiation can be estimated by the determination of bone-like calcium deposits produced on different PVDF composite nanofibers using ARS staining [1,4]. After the incubation for 3, 5, 7, and 14 days, the absorbance changes of ARS in the red-stained composite nanofibers were used to quantify the calcium mineralization. Figure 10b exhibits the absorbance of ARS extracted from the stained calcium deposits in different composite nanofibers. The increase of calcium deposition on the composite nanofibers with time was observed in all samples. Among the samples, PE06 exhibited the highest calcium deposition compared to the other samples at all time points caused by increased osteogenesis capability [42]. These results indicated that the amount of POSS–EGCG conjugate present in the composite nanofibers significantly influenced the bone-forming ability of the MC3T3-E1 cells.

### 3.6. Quantification of Inflammatory Cytokine

Osteoblasts can produce bone-active cytokines such as tumor necrosis factor-α and IL-6 [21,38]. In bone metabolism, these cytokines act as potent factors for osteoclast formation and bone resorption because they can mediate the effects of many stimulators of bone resorption, i.e., parathyroid hormone and IL-1 [22,23,40]. Among these cytokines, IL-6—the most potent osteoclastogenic factor—promotes the activation of osteoclast precursors and their subsequent differentiation into mature osteoclasts, leading to the bone resorption. In addition, a significant release of IL-6 can suppress cell viability. Thus, we determined the expression of IL-6 in cell cultures to evaluate the bone-forming ability and cytocompatibility of the PVDF composite nanofibers.

It was found that PVDF and PO06 promoted the expression of the IL-6 markedly due to the contamination by LPS and subsequent macrophage activation (Figure 11). However, all composite nanofibers containing the POSS–EGCG conjugate exhibited considerably decreased levels of IL-6 expression compared with the pure PVDF nanofiber. In particular, PE06 exhibited only a very small amount of IL-6 secretion, suggesting that the POSS–EGCG conjugate can downregulate the production of bone-active cytokines and improve the cytocompatibility of the composite nanofibers via the anti-inflammatory effect of EGCG [22,23,40]. Therefore, the content of the POSS–EGCG conjugate present was an essential factor in decreasing IL-6 production on the PVDF composite nanofibers.

## 4. Conclusions

In this study, we fabricated PVDF composite nanofibers as potential membranes for bone tissue regeneration using an electrospinning process. The composite nanofibers obtained had fully interconnected porous structures adequate to promote the cell proliferation and the efficient transport of nutrients and metabolic waste. The content of POSS–EGCG conjugate present affected the piezoelectric, antioxidant, and anti-inflammatory properties of the nanofibers, which finally influenced the in vitro bioactivity of the PVDF composite nanofibers. PE06 facilitated the proliferation and differentiation of the MC3T3-E1 cells more effectively compared to the other nanofibers. Based on these results, our simple method is very useful to prepare PVDF composite nanofibers which may contribute to the development of new scaffolding materials for bone tissue regeneration.

## Figures and Tables

**Figure 1 nanomaterials-09-00184-f001:**
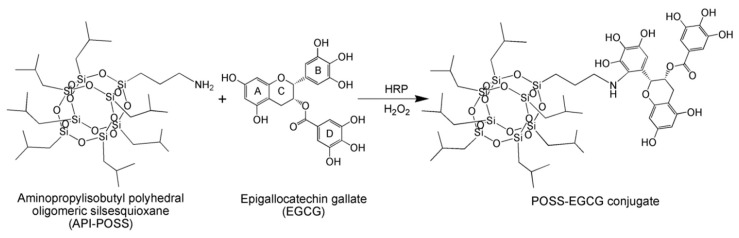
Schematic diagram of enzymatic synthesis of the polyhedral oligomeric silsesquioxane–epigallocatechin gallate (POSS–EGCG) conjugate.

**Figure 2 nanomaterials-09-00184-f002:**
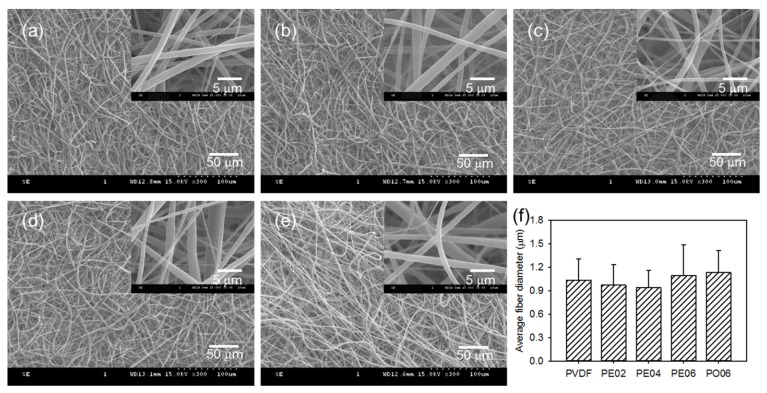
SEM images of (**a**) PVDF, (**b**) PE02, (**c**) PE04, (**d**) PE06, and (**e**) PO06 composite nanofibers. (**f**) Average fiber diameters of the PVDF composite nanofibers analyzed from the SEM images (*n* = 4).

**Figure 3 nanomaterials-09-00184-f003:**
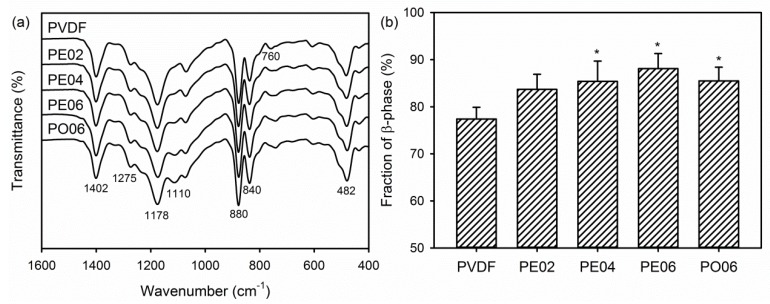
(**a**) FTIR spectra of the PVDF composite nanofibers and (**b**) changes in β-phase content in the nanofibers calculated from the FTIR spectra (*n* = 5). Significant difference from the pure PVDF nanofiber was denoted as *p** ˂ 0.05.

**Figure 4 nanomaterials-09-00184-f004:**
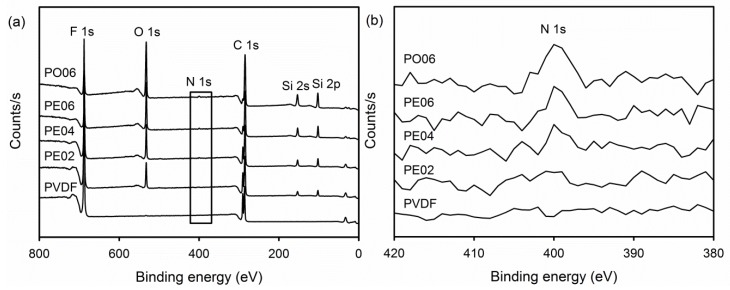
(**a**) X-ray photoelectron (XPS) spectra of the PVDF composite nanofibers and (**b**) the magnified XPS spectra of the areas in (a) marked by rectangle.

**Figure 5 nanomaterials-09-00184-f005:**
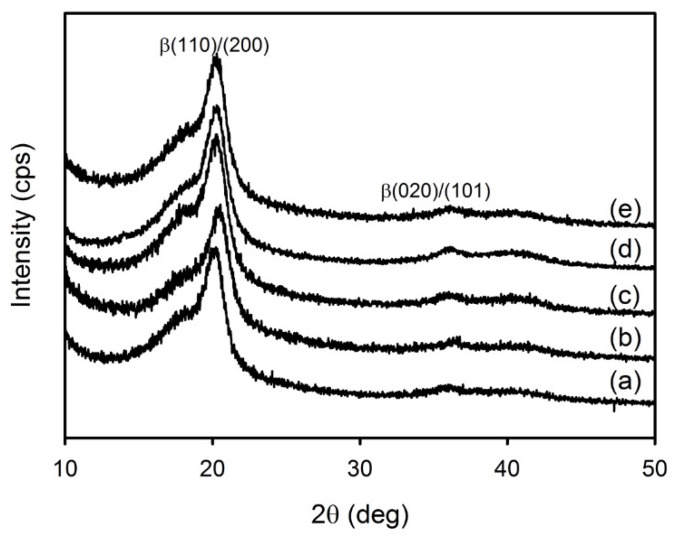
XRD patterns of (a) PVDF, (b) PE02, (c) PE04, (d) PE06, and (e) PO06 composite nanofibers.

**Figure 6 nanomaterials-09-00184-f006:**
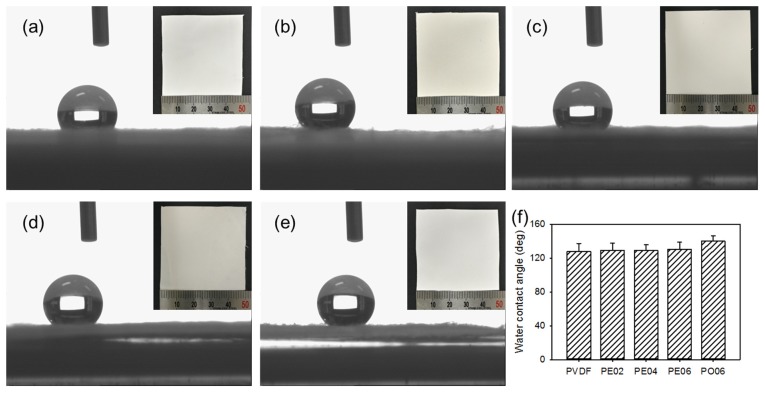
Contact angle images for water droplet on (**a**) PVDF, (**b**) PE02, (**c**) PE04, (**d**) PE06, and (**e**) PO06 composite nanofibers. (**f**) Average contact angles on PVDF composite nanofibers determined by contact angle meter (*n* = 5). Insets are photographs of PVDF composite nanofibers.

**Figure 7 nanomaterials-09-00184-f007:**
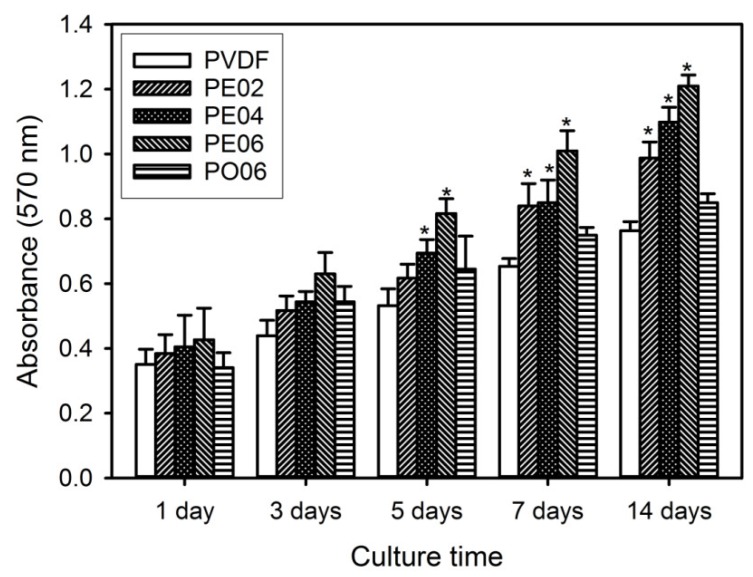
Proliferation behavior of MC3T3-E1 cells cultured on the PVDF composite nanofibers (*n* = 6). Significant difference from the pure PVDF nanofiber at each time point was denoted as *p** ˂ 0.05.

**Figure 8 nanomaterials-09-00184-f008:**
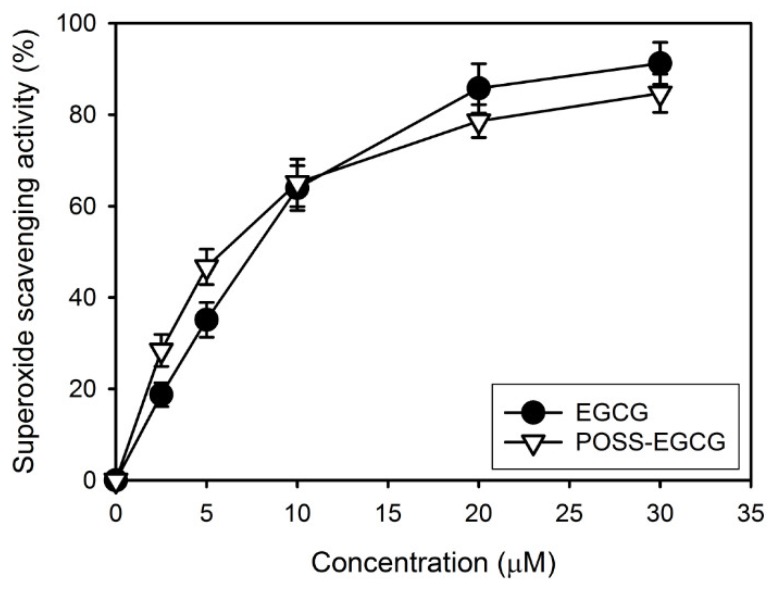
Superoxide anion scavenging activities of EGCG and the POSS–EGCG conjugate (*n* = 4).

**Figure 9 nanomaterials-09-00184-f009:**
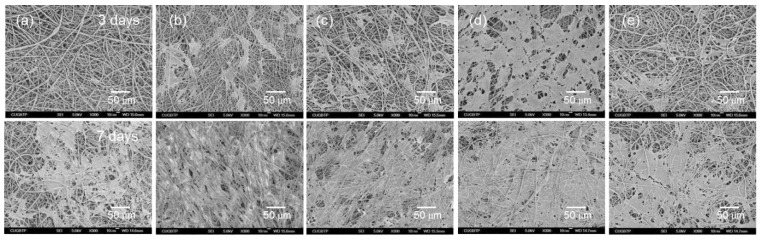
Morphologies of MC3T3-E1 cells on (**a**) PVDF, (**b**) PE02, (**c**) PE04, (**d**) PE06, and (**e**) PO06 composite nanofibers after culturing for 3 and 7 days.

**Figure 10 nanomaterials-09-00184-f010:**
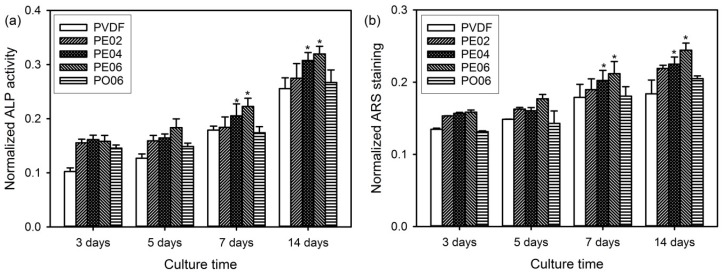
(**a**) ALP activity and (**b**) quantitative calcium deposition of MC3T3-E1 cells on the PVDF composite nanofibers after culturing for 3, 5, 7, and 14 days (*n* = 6). The values of ALP activity and quantitative calcium deposition were normalized by the cell number (1 × 10^4^ cells). Significant difference from the pure PVDF nanofiber at each time point was denoted as *p** ˂ 0.05.

**Figure 11 nanomaterials-09-00184-f011:**
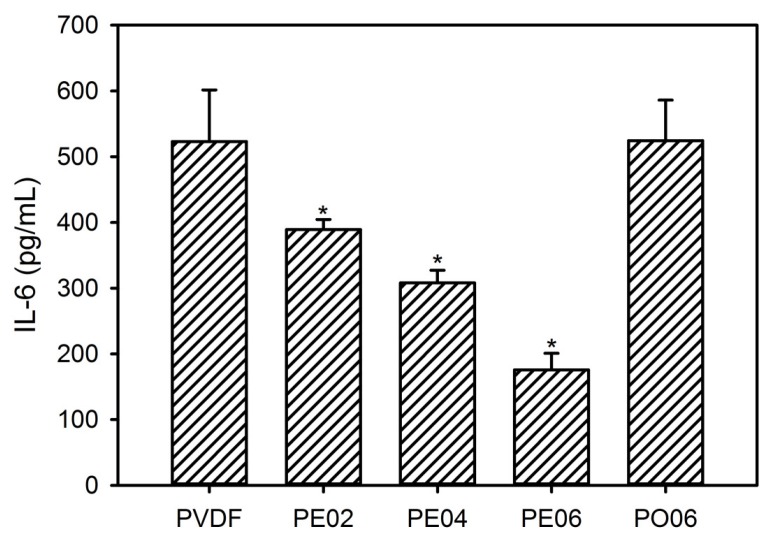
Levels of IL-6 expression by RAW 264.7 cells on the PVDF composite nanofibers (*n* = 5). Significant difference from the pure PVDF nanofiber at each time point was denoted as *p** ˂ 0.05.

**Table 1 nanomaterials-09-00184-t001:** Mechanical properties of the PVDF composite nanofibers with different contents of the POSS–EGCG conjugate.

Sample	Content of POSS–EGCG Conjugate (wt%)	Young’s Modulus (MPa)	Tensile Strength (MPa)	Elongation at Break (%)
PVDF	0	3.5 ± 0.3	1.1 ± 0.1	72.6 ± 6.0
PE02	2	3.6 ± 0.2	1.1 ± 0.3	71.3 ± 6.5
PE04	4	4.0 ± 0.3	1.4 ± 0.2	69.1 ± 7.7
PE06	6	4.5 ± 0.5	1.6 ± 0.3	65.6 ± 7.2
PO06	6^a^	5.2 ± 0.4	1.7 ± 0.3	63.4 ± 8.4

^a^ Content of API-POSS (wt%).

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
