# Peer review of "Poly(vinylidene fluoride) Composite Nanofibers Containing Polyhedral Oligomeric Silsesquioxane–Epigallocatechin Gallate Conjugate for Bone Tissue Regeneration"

_nanomaterials, 2019, doi:10.3390/nano9020184_

Reviewer 1 Report

In the manuscript "Poly(vinylidene fluoride) Composite Nanofibers Containing Polyhedral Oligomeric Silsesquioxane–Epigallocatechin Gallate Conjugate for Bone Tissue Regeneration", authors prepared, characterized and in vitro tested piezoelectric antioxidant substrates for bone tissue engineering. The article is scientifically interesting and well written. However, the following revisions are required for the publication of the manuscript.

-Lines 149-151; please explain if and how the samples were fixed to the bottom of the wells.

-Section 3.1 (Synthesis of the POSS–EGCG conjugate); please, show the NMR spectra related to both POSS and EGCG in order to compare their characteristic speaks to the POSS-EGCG conjugate.

-Line 243; please remove the sentence "The FTIR was used to analysis the chemical structure" and substitute it with "The FTIR was used to analyze the chemical structure"

-Lines 260-261: "In addition, the piezoelectric β-phase increased from 77.4% to 88.1% after adding the POSS–EGCG conjugate"; please, write also the standard deviation values in the text; is this difference significant? Perform statistic analysis and, in case, show the asterisk in the relative graph (Figure 3b).

-All the above considerations are also applied to the lines 293-295: "In addition, the water contact angles were hardly changed by the addition of the POSS–EGCG conjugate, i.e., 129.0° for PE02, 128.9° for PE04, and 130.4° for PE06."

-Introduction requires to be improved; from line 48 to 59, piezoelectricity and PVDF are presented in the text without specifying the importance of the piezoelectric scaffolds in bone tissue engineering and that the bone itself is a piezoelectric material. Please, expand this part and report the following references in this regard: DOI1: 10.1016/j.actbio.2015.07.010; DOI2: 10.1021/acsami.7b04323; DOI3: 10.1016/j.nano.2017.05.006; DOI4: 10.1021/acsami.5b08764;

Author Response

Reviewer 1

In the manuscript "Poly(vinylidene fluoride) Composite Nanofibers Containing Polyhedral Oligomeric Silsesquioxane–Epigallocatechin Gallate Conjugate for Bone Tissue Regeneration", authors prepared, characterized and in vitro tested piezoelectric antioxidant substrates for bone tissue engineering. The article is scientifically interesting and well written. However, the following revisions are required for the publication of the manuscript.

Point 1: -Lines 149-151; please explain if and how the samples were fixed to the bottom of the wells.

Response 1: According to the comment of the reviewer, we inserted on sentence to explain the fixation of the samples into the bottom of the well in page 4, line 161. We put circular-shaped nanofibers into 24-well culture plate and fixed nanofibers with glass ring.

Point 2: -Section 3.1 (Synthesis of the POSS–EGCG conjugate); please, show the NMR spectra related to both POSS and EGCG in order to compare their characteristic speaks to the POSS-EGCG conjugate.

Response 2: As commented by the reviewer, we added some sentences to explain the structures of EGCG and API-POSS determined by 1H NMR spectroscopy in page 3, lines 111-115. In addition, 1H NMR spectra of EGCG and API-POSS were added in Figure S1.

Point 3: -Line 243; please remove the sentence "The FTIR was used to analysis the chemical structure" and substitute it with "The FTIR was used to analyze the chemical structure"

Response 3: As pointed out by the reviewer, we removed the sentence “The FTIR was used to analysis the chemical structure” and substituted it with “The FTIR was used to analyze the chemical structure”. This is our mistake. Thank you for kind advice.

Point 4: -Lines 260-261: "In addition, the piezoelectric β-phase increased from 77.4% to 88.1% after adding the POSS–EGCG conjugate"; please, write also the standard deviation values in the text; is this difference significant? Perform statistic analysis and, in case, show the asterisk in the relative graph (Figure 3b).

Response 4: As commented by the reviewer, we added the standard deviation values of β-phase content in the PVDF composite nanofibers in page 7, line 274. In addition, we performed the statistical analysis of the result of β-phase content determination and added significance difference using asterisk in Figure 3b

Point 5: -All the above considerations are also applied to the lines 293-295: "In addition, the water contact angles were hardly changed by the addition of the POSS–EGCG conjugate, i.e., 129.0° for PE02, 128.9° for PE04, and 130.4° for PE06."

Response 5: As commented by the reviewer, we added the standard deviation values of contact angles of the PVDF composite nanofibers in page 8, line 308. We also performed the statistical analysis of the result of contact angle measurement, but significance difference was not observed.

Point 6: -Introduction requires to be improved; from line 48 to 59, piezoelectricity and PVDF are presented in the text without specifying the importance of the piezoelectric scaffolds in bone tissue engineering and that the bone itself is a piezoelectric material. Please, expand this part and report the following references in this regard: DOI1: 10.1016/j.actbio.2015.07.010; DOI2: 10.1021/acsami.7b04323; DOI3: 10.1016/j.nano.2017.05.006; DOI4: 10.1021/acsami.5b08764;

Response 6: According to the comment of the reviewer, we added some sentences to specify the importance of the piezoelectric scaffolds in bone tissue engineering in page 2, lines 61-63. We also added reference related to the piezoelectric scaffolds. Thank you for kind advice.

Reviewer 2 Report

I attached reviewer comment.

Author Response

Reviewer 2

Thank you for inviting as a reviewer of original manuscript entitled “Poly(vinylidene fluoride) Composite Nanofibers Containing Polyhedral Oligomeric Silsesquioxane–Epigallocatechin Gallate Conjugate for Bone Tissue Regeneration” by Jeong et al. submitted to Nanomaterials. I carefully reviewed it. The overall research quality and scope of this study is excellent and suitable for publication. The experiments are well equipped, and the conclusion are well suspended by their results. Of course, I found several correction points for improving this manuscript. I listed below.

Point 1: Major point

Although authors showed good resulting data, I found no photographs of surface of samples for contact angle evaluation. In my opinion, adding photographs of samples or low magnified SEM micrographs of samples (such as Fig.9) are enhancing the quality of this manuscript.

Response 1: As commented by the reviewer, we added photographs of the composite nanofibers in Figure 6 (contact angle images). In addition, we added low magnification SEM images in Figure 2 for showing the surface of the composite nanofibers.

Point 2: Abstract

In P1L17, electrospinning →electrospinning methods.

Response 2: As pointed out by the reviewer, we changed sentence from “electrospinning” to “electrospinning methods”.

Point 3: Introduction

Point 3-1: In P2L44, add below references to [4-6]. Acta Biomater., 10[8], 3733-3746 (2014); Appl. Surface Sci., 309, 231-239 (2014); RSC Adv., 4, 52491-52499 (2014); Biomaterials, 27, 1216-1222 (2006)

Response 3-1: According to the comment of the reviewer, we added four references in the text such as ref. 6, ref. 8, ref. 9, and ref. 10. Thank you for good advice.

Point 3-2: In P2L49, add a sentence about the details of osteoconduction.

Response 3-2: As pointed out by the reviewer, we added a sentence to explain the osteoconduction in page 1, line 37.

Point 4: Materials and Methods

In P3L105, revise Bruker. Which Bruker? AXS?

Response 4: Bruker BioSpin is a manufacturer of NMR and thus we added the company name, Bruker BioSpin, in page 3, line 110.

Point 5: Results and Discussions

Point 5-1: In Fig. 1, is it polymer? if so, add ( materials )n and residual product in this chemical formula scheme.

Response 5-1: POSS-EGCG conjugate is not a polymer and its molecular weight is about 1,330. After the synthesis of the POSS-EGCG conjugate, the resulting product was dialyzed in DW for 48, followed by centrifugation and washing with DW to eliminate the residual reactants.

Point 5-2: In Fig. 4, XPS survey scan spectra →XPS spectra. In my opinion, it is better way to add expanded spectra of N1s as fig. 4(b).

Response 5-2: According to the comment of the reviewer, we changed sentence from “XPS survey scan spectra” to “XPS spectra” in the caption of Figure 4. In addition, magnified N 1s spectra in Figure 4 were changed to expanded N 1S spectra as Figure 4b. Thank you for kind advice.

Round  2

Reviewer 1 Report

The authors successfully addressed the revisions; I recommend to accept the article for publication.